# The drift diffusion model as the choice rule in inter-temporal and risky choice: A case study in medial orbitofrontal cortex lesion patients and controls

**Jan Peters**[1]*, **Mark D'Esposito**[2]

**1** Department of Psychology, Biological Psychology, University of Cologne, Germany, **2** Helen Wills Neuroscience Institute, University of California at Berkeley, Berkeley, California, United States of America

* jan.peters@uni-koeln.de

**Data Availability Statement:** Data cannot be shared publicly because participants did not provide consent for having the data posted in a public repository. Data are available from https://

## Abstract

Sequential sampling models such as the drift diffusion model (DDM) have a long tradition in research on perceptual decision-making, but mounting evidence suggests that these models can account for response time (RT) distributions that arise during reinforcement learning and value-based decision-making. Building on this previous work, we implemented the DDM as the choice rule in inter-temporal choice (temporal discounting) and risky choice (probability discounting) using hierarchical Bayesian parameter estimation. We validated our approach in data from nine patients with focal lesions to the ventromedial prefrontal cortex / medial orbitofrontal cortex (vmPFC/mOFC) and nineteen age- and education-matched controls. Model comparison revealed that, for both tasks, the data were best accounted for by a variant of the drift diffusion model including a non-linear mapping from value-differences to trial-wise drift rates. Posterior predictive checks confirmed that this model provided a superior account of the relationship between value and RT. We then applied this modeling framework and 1) reproduced our previous results regarding temporal discounting in vmPFC/mOFC patients and 2) showed in a previously unpublished data set on risky choice that vmPFC/mOFC patients exhibit increased risk-taking relative to controls. Analyses of DDM parameters revealed that patients showed substantially increased non-decision times and reduced response caution during risky choice. In contrast, vmPFC/mOFC damage abolished neither scaling nor asymptote of the drift rate. Relatively intact value processing was also confirmed using DDM mixture models, which revealed that in both groups >98% of trials were better accounted for by a DDM with value modulation than by a null model without value modulation. Our results highlight that novel insights can be gained from applying sequential sampling models in studies of inter-temporal and risky decision-making in cognitive neuroscience.

zenodo.org/record/3742412 for researchers who meet the criteria for access to confidential data.

**Funding:** This work was funded by Deutsche Forschungsgemeinschaft (grants PE 1627/4-1 and PE1627/5-1 to J.P.). The funders had no role in study design, data collection and analysis, decision to publish, or preparation of the manuscript.

**Competing interests:** The authors have declared that no competing interests exist.

## Author summary

Maladaptive changes in decision-making are associated with many psychiatric and neurological disorders, e.g. when people are making impulsive or risky decisions. For understanding the processes of how such decisions arise, it can be informative to examine not only the choices that people make, but also the response times associated with these decisions. Here we show that response times during impulsive and risky decision-making are well accounted for by a model that has been developed to describe perceptual decision-making, the drift diffusion model. Furthermore, we use this model to examine impulsive and risky choice following damage to a core regions of the brains decision-making circuitry, the ventromedial / orbitofrontal cortex. Although this region has repeatedly been shown to contribute to value processing, modeling revealed that lesions to this area do not render reponse times less dependent on value. Our results highlight that novel insights can be gained from applying such models in studies of impulsive and risky choice in cognitive neuroscience.

## Introduction

Understanding the neuro-cognitive mechanisms underlying decision-making and reinforcement learning[1–3] has potential implications for many neurological and psychiatric disorders associated with maladaptive choice behavior[4–6]. Modeling work in value-based decision-making and reinforcement learning often relies on simple logistic (softmax) functions[7,8] to link model-based decision values to observed choices. In contrast, in perceptual decision-making, sequential sampling models such as the drift diffusion model (DDM) that not only account for the observed choices but also for the full response time (RT) distributions have a long tradition[9–11]. Recent work in reinforcement learning[12–15], inter-temporal choice [16,17] and value-based choice[18–21] has shown that sequential sampling models can be successfully applied in these domains.

In the DDM, decisions arise from a noisy evidence accumulation process that terminates as the accumulated evidence reaches one of two response boundaries[9]. In its simplest form, the DDM has four free parameters: the boundary separation parameter $\alpha$ governs how much evidence is required before committing to a decision. The upper boundary corresponds to the case when the accumulated evidence exceeds $\alpha$, whereas the lower boundary corresponds to the case when the accumulated evidence exceeds zero. The drift rate parameter $v$ determines the mean rate of evidence accumulation. A greater drift rate reflects a greater rate of evidence accumulation and thus faster and more accurate responding. In contrast, a drift rate of zero would indicate chance level performance, as the evidence accumulation process would have an equal likelihood of terminating at the upper or lower boundaries (for a neutral bias). The starting point or bias parameter $z$ determines the starting point of the evidence accumulation process in units of the boundary separation, and the non-decision time $\tau$ reflects components of the RT related to stimulus encoding and/or response preparation that are unrelated to the evidence accumulation process. The DDM can account for a wide range of experimental effects on RT distributions during two-alternative forced choice tasks[9].

The application of sequential sampling models such as the DDM has several potential advantages over traditional softmax[7] choice rules. First, including RT data during model estimation may improve both the reliability of the estimated parameters[12] and parameter recovery[13], thereby leading to more robust estimates. Second, taking into account the full RT distributions can reveal additional information regarding the dynamics of decision

processes[14,15]. This is of potential interest, in particular in the context of maladaptive behaviors in clinical populations[14,22–25] but also when the goal is to more fully account for how decisions arise on a neural level[10].

In the present case study, we focus on a brain region that has long been implicated in decision-making, reward-based learning and impulse regulation[26,27], the ventromedial prefrontal / medial orbitofrontal cortex (vmPFC/mOFC). Performance impairments on the Iowa Gambling Task are well replicated in vmPFC/mOFC patients[26,28,29]. Damage to vmPFC/mOFC also increases temporal discounting[30,31] (but see[32]) and risk-taking[33–35], impairs reward-based learning[36–38] and has been linked to inconsistent choice behavior [39–41]. Meta-analyses of functional neuroimaging studies strongly implicate this region in reward valuation[42,43]. Based on these observations, we reasoned that vmPFC/mOFC damage might also render RTs during decision-making less dependent on value. In the context of the DDM, this could be reflected in changes in the value-dependency of the drift rate $v$. In contrast, more general impairments in the processing of decision options, response execution and/or preparation would be reflected in changes in the non-decision time. Interestingly, however, one previous model-free analysis in vmPFC/mOFC patients revealed a similar modulation of RTs by value in patients and controls[40].

The present study therefore had the following aims. The first aim was a validation of the applicability of the DDM as a choice rule in the context of inter-temporal and risky choice. To this end, we first performed a model comparison of variants of the DDM in a data set of nine vmPFC/mOFC lesion patients and nineteen controls. Since recent work on reinforcement learning suggested that the mapping from value differences to trial-wise drift rates might be non-linear[15] rather than linear[14], we compared these different variants of the DDM in our data and ran posterior predictive checks on the winning DDM models to explore the degree to which the different models could account for RT distributions and the relationship between RTs and subjective value. Second, we re-analyzed previously published temporal discounting data in controls and vmPFC/mOFC lesion patients to examine the degree to which our previously reported model-free analyses[30] could be reproduced using a hierarchical Bayesian model-based analysis with the DDM as the choice rule. Third, we used the same modeling framework to analyze previously unpublished data from a risky decision-making task in the same lesion patients and controls to examine whether risk taking in the absence of a learning requirement is increased following vmPFC/mOFC damage. Finally, we explored changes in choice dynamics as revealed by DDM parameters as a result of vmPFC/mOFC lesions, and investigated whether lesions to vmPFC/mPFC impacted the degree to which RTs were sensitive to subjective value differences, both by examining DDM parameters and via DDM mixture models.

## Results

### Model comparison

We first compared the fit of two previously proposed DDM models with linear (DDM$_{lin}$, see Eq 5)[14] and non-linear (DDM$_S$, see Eq 6 and Eq 7)[15] value-dependent drift-rate scaling in terms of the WAIC and the estimated log predictive density (elpd)[44]. For comparison we also included a null model (DDM$_0$) with constant drift rate, that is, a model without value modulation. For both temporal discounting data (Table 1) and risky choice / probability discounting data (Table 2), the non-linear drift rate scaling models outperformed linear scaling, and both models fit the data better than the DDM$_0$. Furthermore, 95% confidence intervals of the differences in elpd between each model and the DDM$_S$ did not overlap, and did not include 0 (Tables 1 and 2, last column), suggesting that the differences in elpd were robust.

**Table 1. Model comparison of drift diffusion models of temporal discounting.** The hyperbolic+shift value function (see Eq 1) corresponds to hyperbolic discounting in the *now* condition, and a shift parameter that models the decrease in discounting between the *now* and *not now* conditions. WAIC–Widely Applicable Information Criterion; elpd–estimated log predictive density; $elpd_{diff}$ is the difference in elpd between each model and the $DDM_S$.

| Model | Drift rate scaling | Value function | WAIC | -elpd | -eldp_diff [95% CI] |
|---|---|---|---|---|---|
| $DDM_0$ | - | - | 20939 | 10472.5 | 1987.9 [1899.1–2076.7] |
| $DDM_{lin}$ | Linear | Hyperbolic+Shift | 19602 | 9805.2 | 1320.6 [1231.2–1409.9] |
| $DDM_S$ | Sigmoid | Hyperbolic+Shift | 16966 | 8484.6 | - |

## Model validation

We then carried out a number of simple sanity checks (see S1 Text) which confirmed that log (k) parameters estimated via standard softmax and via the $DDM_s$ showed good correspondence (S3 Fig). Likewise, minimum and median RT showed the expected associations with model-based non-decision times (S4 Fig) and boundary separation parameters (S5 Fig).

## Prediction of binary choice data

We then checked the degree to which the different implementations of the DDM predicted participants' binary choices. Using each participant's mean posterior parameters from the hierarchical models we calculated model predicted choices, and compared these to the observed binary choices. Raw accuracy scores per model and group are listed in Table 3 (temporal discounting) and Table 4 (risky choice) with the softmax models shown for comparison. Numerically, accuracy scores for the $DDM_S$ were higher than for $DDM_{lin}$. Indeed variance-stabilized accuracy values (arcsine-square-root transformed, see Fig 1) were greater for $DDM_S$ compared to $DDM_{lin}$ for temporal discounting ($t_{27}$ = -7.43, 95% CI: [-.19, -.11]), with a similar trend for risky choice ($t_{27}$ = -1.97, 95% CI: [-.09, .002]).

## Posterior predictive checks and prediction of RTs

Next, we carried out posterior predictive checks (see methods section) to 1) examine whether models also differed with respect to their ability to account the observed RTs (as opposed to only binary choices) and 2) to verify that the best-fitting model captured the overall pattern in the data. Posterior predictive checks for the $DDM_S$ for each individual participant in relation to the full RT distributions are shown in the SI for temporal discounting (S1 Fig) and risky choice (S2 Fig). These initial checks revealed that the $DDM_S$ indeed provided a good account of individual RT distributions.

In a second step, we directly compared the ability of the $DDM_S$ and $DDM_{lin}$ to account for how value modulates RTs. To this end, we binned trials for each subject into five bins according to the subjective value of the LL or risky reward according to Eqs 1 and 2. We then simulated 10k full data sets from the posterior distributions of each model ($DDM_0$, $DDM_{lin}$, $DDM_S$) and averaged model predicted response times per bin. Results are shown for each participant in Fig 2 for temporal discounting and Fig 3 for risky choice. The $DDM_0$ does not

**Table 2. Model comparison of drift diffusion models of risky choice.** The hyperbolic value function (see Eq 2) corresponds to hyperbolic discounting over the odds-against-winning the gamble. WAIC–Widely Applicable Information Criterion; elpd–estimated log predictive density; $elpd_{diff}$ is the difference in elpd between each model and the $DDM_S$.

| Model | Drift rate scaling | Value function | WAIC | -elpd | -eldp_diff [95% CI] |
|---|---|---|---|---|---|
| $DDM_0$ | - | - | 11515 | 5760.3 | 1162.8 [1094.4–1231.2] |
| $DDM_{lin}$ | Linear | Hyperbolic | 10422 | 5222.4 | 625.0 [546.0–703.9] |
| $DDM_S$ | Sigmoid | Hyperbolic | 9190 | 4597.4 | - |

**Table 3. Median (range) of the proportion of correctly predicted binary choices for the different temporal discounting models, separately for mOFC patients and controls.**

|  | Softmax | $DDM_{lin}$ | $DDM_S$ |
|---|---|---|---|
| **mOFC patients** | .92 (.87-.99) | .90 (.80-.96) | .92 (.89-.99) |
| **Controls** | .91 (.78-.96) | .75 (.60-.99) | .91 (.84-.99) |

incorporate values, thus it predicts the same RTs across value bins (horizontal blue lines in Figs 2 and 3). While the $DDM_{lin}$ could account for some aspects of the association between value and RT in some participants, the $DDM_S$ provided a much better account of this relationship overall.

This was in many cases due to the $DDM_{lin}$ overestimating RTs (underestimating the drift rate) for intermediate value trials and underestimating RTs (overestimating the drift rate) for trials with high value LL or risky options. This effect is most clearly seen in the temporal discounting data (Fig 2) where a greater proportion of value bins fall into the intermediate range. In the supplemental information, we visually compare predicted drift rates between $DDM_{lin}$ and $DDM_S$ to illustrate this effect (S6 Fig). Taken together, these analyses show that 1) the $DDM_S$ provided an overall superior fit to both temporal discounting and risky choice data and 2) that this was reflected in a better account of both binary choices and the relationship between RTs and value.

## Simulations of effects of drift rate components on RT distributions

We next set out to more systematically explore how the two components of the drift rate in the $DDM_S$ ($v_{max}$ and $v_{coeff}$) affect RTs. To this end, we simulated 50 RTs from the $DDM_S$ for each of 400 value differences ranging from zero to ± 20. We ran 30 simulations in total, systematically varying $v_{max}$ and $v_{coeff}$ while keeping the other DDM parameters (boundary separation, bias, non-decision time) fixed at mean posterior values of the control group (see Table 5).

Simulated RT distributions are shown in Fig 4A, whereas mean simulated RTs and binary choices per value bin are shown in Fig 4B and 4C, respectively. Results from corresponding simulations computed across the actual delay/amount and probability/amount combinations from the tasks are shown in S8 Fig (temporal discounting) and S9 Fig (risky choice). As can be seen in Fig 4A, the effects of $v_{max}$ on the leading edge of the RT distribution were generally more pronounced for higher values of $v_{coeff}$. At the same time, smaller values of $v_{coeff}$ generally lead to more heavy tailed RT distributions. The model of course predicts longest RTs for trials were values are most similar (the predicted RTs are highest for value differences close to zero, see the dotted lines in the right panels of Fig 4B). But the simulations illustrate an additional effect: Both relatively high and relatively low values of $v_{coeff}$ can make RTs appear insensitive to value differences. For example, for the case of $v_{coeff} = .05$, RTs tend to be uniformly slow, and accelerate only slightly for the largest value differences (blue lines in Fig 4B). In contrast, for the highest values of $v_{coeff}$, relatively small value differences already give rise to maximal drift rates and thus uniformly fast RTs for all but the smallest value differences (highest conflict).

**Table 4. Median (range) of the proportion of correctly predicted binary choices for the different risky choice models, separately for mOFC patients and controls.**

|  | Softmax | $DDM_{lin}$ | $DDM_S$ |
|---|---|---|---|
| **mOFC patients** | .92 (.82-.97) | .90 (.82-.95) | .92 (.84-.98) |
| **Controls** | .92 (.82-.99) | .91 (.79-.99) | .91 (.82-.99) |

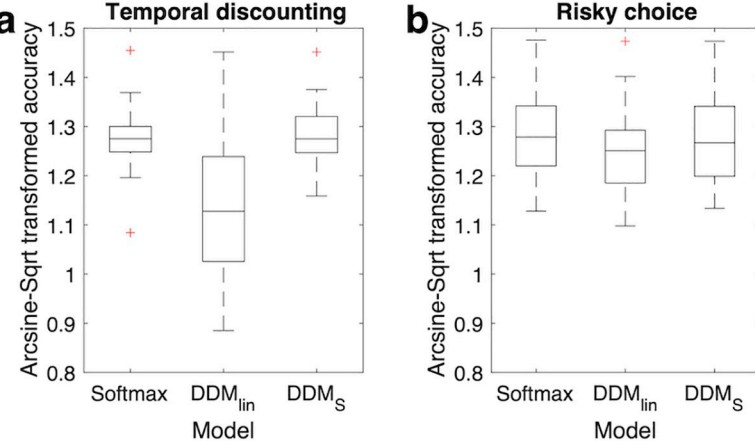

**Fig 1.** Variance-stabilized proportion of trials (arcsine-square root transformed) where each model correctly predicted binary decisions for temporal discounting (a) and risky choice (b).

## Parameter recovery simulations

A further crucial property of a model is that if generating parameters are known, they should be recoverable. As done in previous work[14,15] we therefore carried out parameter recovery analyses for the most complex model ($DDM_S$). Ten simulated data sets were randomly selected (see methods section) and re-fit using the $DDM_S$. We then compared the generating (true) parameter values to the estimated values. Subject-level parameters generally recovered well (Figs 5A and 6A). Group level means and standard deviations (calculated based on the precision) generally also recovered well (Fig 5B–5E, Fig 6B–6E), such that in most cases, the 95% highest density intervals of the estimated posterior distributions included the true generating parameter values. For parameters that showed a high variance (e.g. $v_{coeff}$ and $log(k)_{now}$ in the patient group) the group-level standard deviations tended to be overestimated.

## Comparison to previous model-free analyses in mOFC patients

We have previously reported that temporal discounting in mOFC lesion patients is more affected by the immediacy of smaller-sooner (SS) rewards than in controls[30]. Our previous analysis revealed this both via an analysis of the area-under-the-curve of the empirical discounting function[45] and by a direct comparison of preference reversals between groups. To further validate the applicability of the DDM in the context of temporal discounting, we next examined whether these effects could be reproduced via the hierarchical $DDM_S$. Fig 7 shows the group-level posterior distributions of parameter means for all seven parameters, where we for the purposes of comparison to our previous results first focus on $log(k)_{now}$ (the discount rate in the baseline *now* condition, see Fig 7F) and $shift_{log(k)}$ (the parameter modeling the *decrease* in discounting in *not now* trials as compared to *now* trials, see Fig 7G). The analysis of directional between-subject effects revealed a numerical increase in $log(k)_{now}$ in the mOFC patient group (Fig 7F, Table 6) and strong evidence for a substantially greater difference in discounting between *now* and *not now* trials in the patients (Fig 7G, Table 6). This shows that our results based on model-free summary measures of discounting behavior following mOFC lesions[30] could be reproduced via a hierarchical Bayesian estimation scheme with the $DDM_S$ as the choice rule.

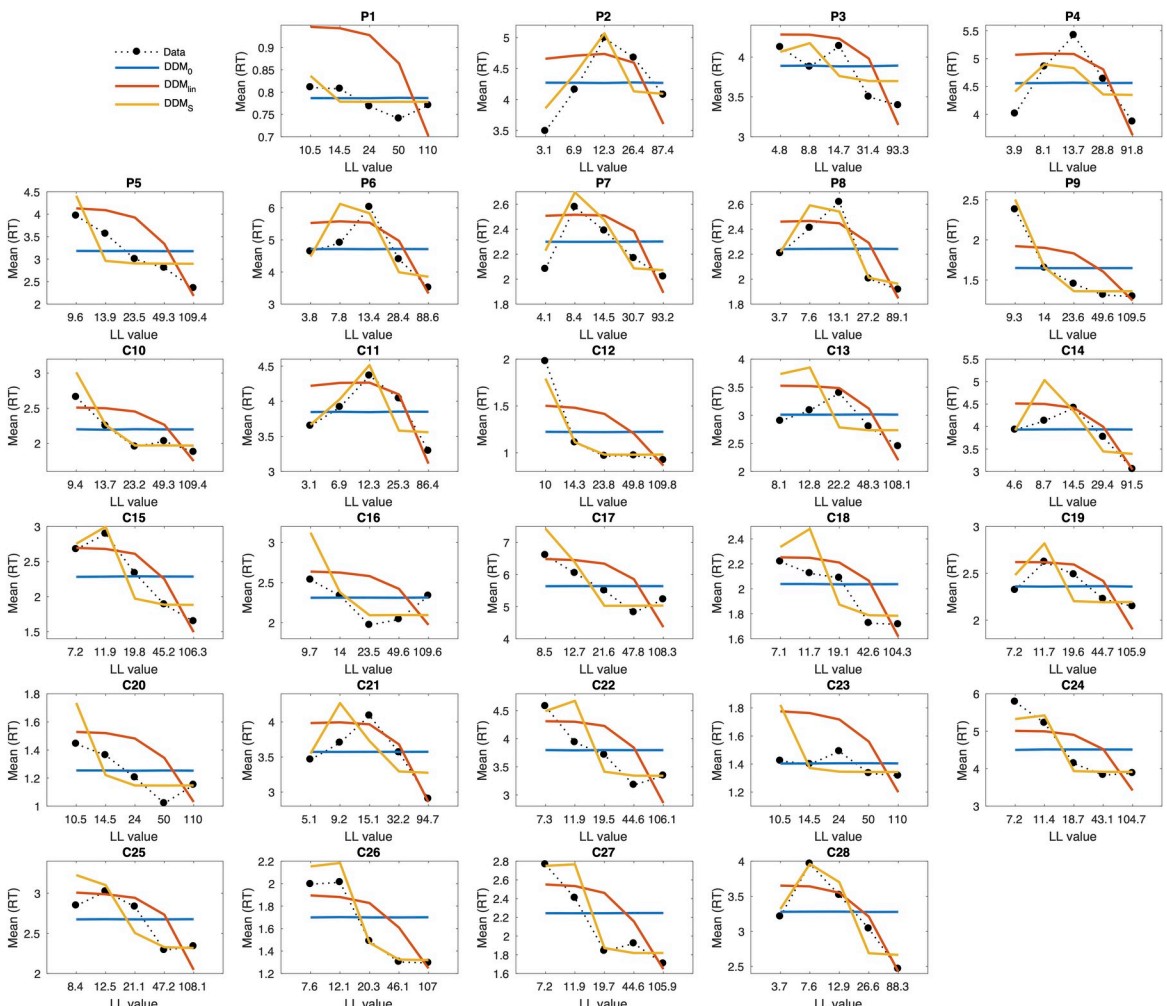

**Fig 2. Posterior predictive plots for the different temporal discounting DDM models for all individual participants (P–mOFC patients, C–controls).** Trials were binned into five bins of equal sizes according to the subjective value of the larger-later (LL) option for each participant (calculated according to Eq 1). The x-axis in each panel shows the subject-specific mean LL value for each bin. The y-axis denotes observed response times per bin (dotted black lines) and model predicted response times per bin for the different DDM models (blue: $DDM_0$, red: $DDM_{lin}$, orange: $DDM_S$). Model predicted response times were obtained by averaging over 10k data sets simulated from the posterior distribution of each hierarchical model.

## Risk-taking in vmPFC/mOFC patients

Risk-taking on the probability discounting task was quantified via the probability discounting parameter *log(h)*, where higher values indicate a greater discounting of value over probabilities. There was some evidence for a smaller *log(h)* in vmPFC/mOFC patients (Fig 8F, Table 6), reflecting a relative increase in risk-taking (reduced value discounting over probabilities) as compared to controls.

## Effects of mOFC lesions on diffusion model parameters

Finally, we examined the diffusion model parameters of the $DDM_S$ models in greater detail. First, there was evidence for longer non-decision times in the patient group for both tasks (see Table 6 and Figs 7B and 8B). These effects amounted to on average 184ms for temporal discounting and 166ms for risky choice. Second, the group differences observed for the starting

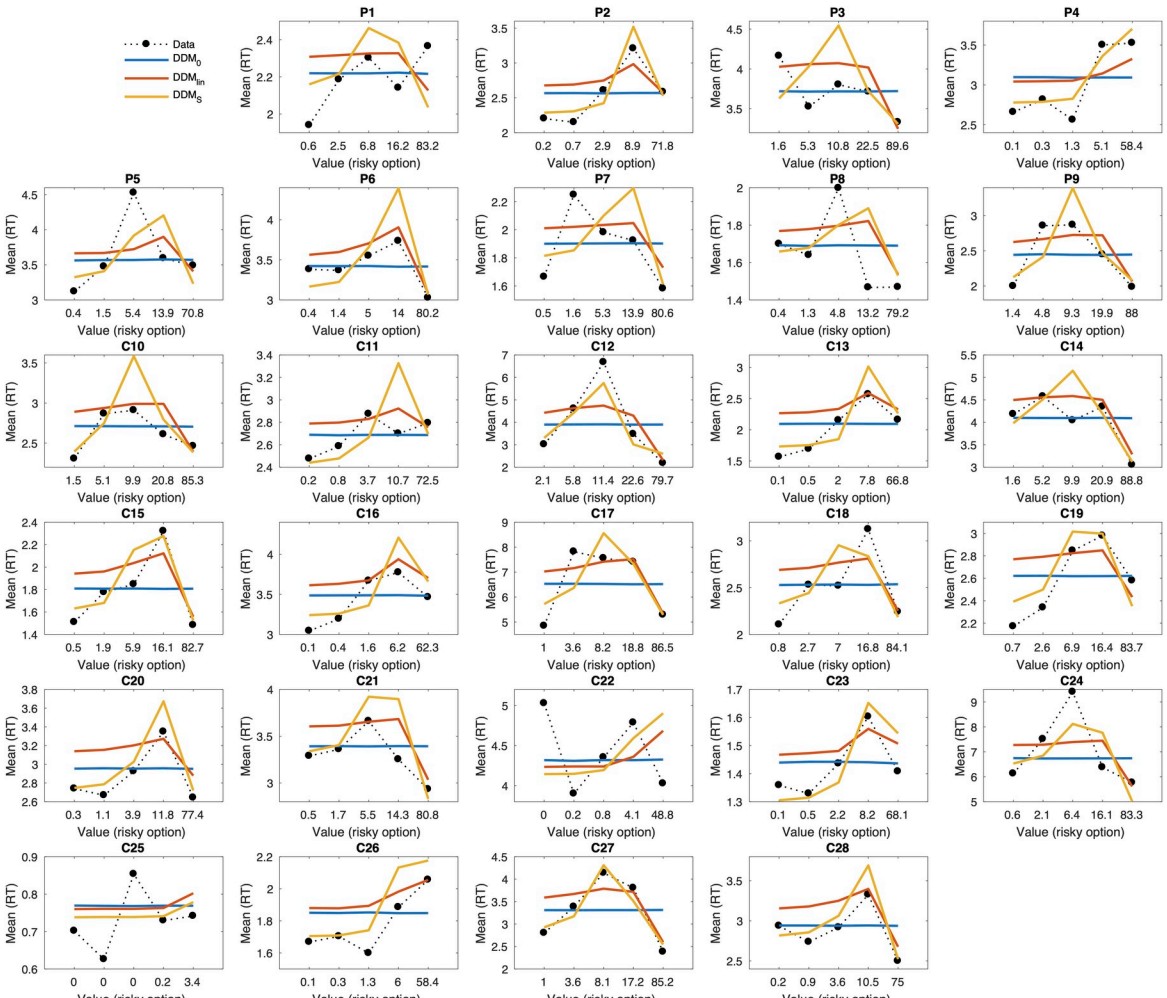

**Fig 3. Posterior predictive plots for the different risky choice DDM models for all individual participants (P–mOFC patients, C–controls).** Trials were binned into five bins of equal sizes according to the subjective value of the risky option for each participant (calculated according to Eq 2). The x-axis in each panel shows the subject-specific mean LL value for each bin. The y-axis denotes observed response times per bin (dotted black lines) and model predicted response times per bin for the different DDM models (blue: $DDM_0$, red: $DDM_{lin}$, orange: $DDM_S$). Model predicted response times were obtained by averaging over 10k data sets simulated from the posterior distribution of each hierarchical model.

point (bias) parameter largely mirrored group differences observed for discounting behavior. For temporal discounting, controls exhibited a more pronounced bias towards the LL boundary than vmPFC/mOFC patients, who exhibited a largely neutral bias here. For risky choice,

**Table 5. DDM parameter values used for simulation analyses depicted in Fig 5.** All parameters are the posterior group means of the control group.

| | Parameter value |
|---|---|
| **Boundary separation ($\alpha$)** | 3.37 |
| **Non decision time ($\tau$)** | .945 |
| **Starting point / bias ($z$)** | .531 |
| **Drift rate $v$ (max)** | [.5, 1, 1.5, 2.5, 3.5] |
| **Drift rate $v$ (coeff)** | [.05, .1, .2, .4, 1, 2] |

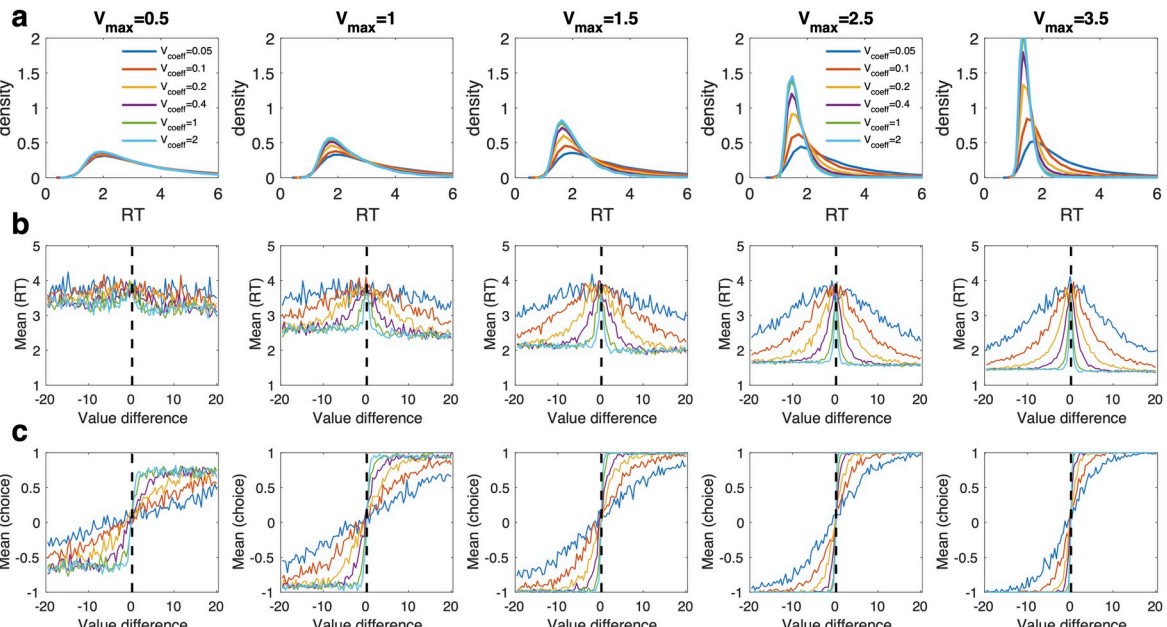

**Fig 4. Simulation results for the DDM$_S$.** a: Simulated RT distributions, b: Predicted mean RTs per value difference bin, c: predicted choice proportions per value difference bin. Simulation results are shown for a range of values of $v_{max}$ (columns) and $v_{coeff}$ (colored lines).

controls showed a bias that was numerically shifted towards the safe option compared to vmPFC/mOFC patients. Third, posterior distributions for the boundary separation parameter (alpha) in temporal discounting showed high overlap and the difference distribution was

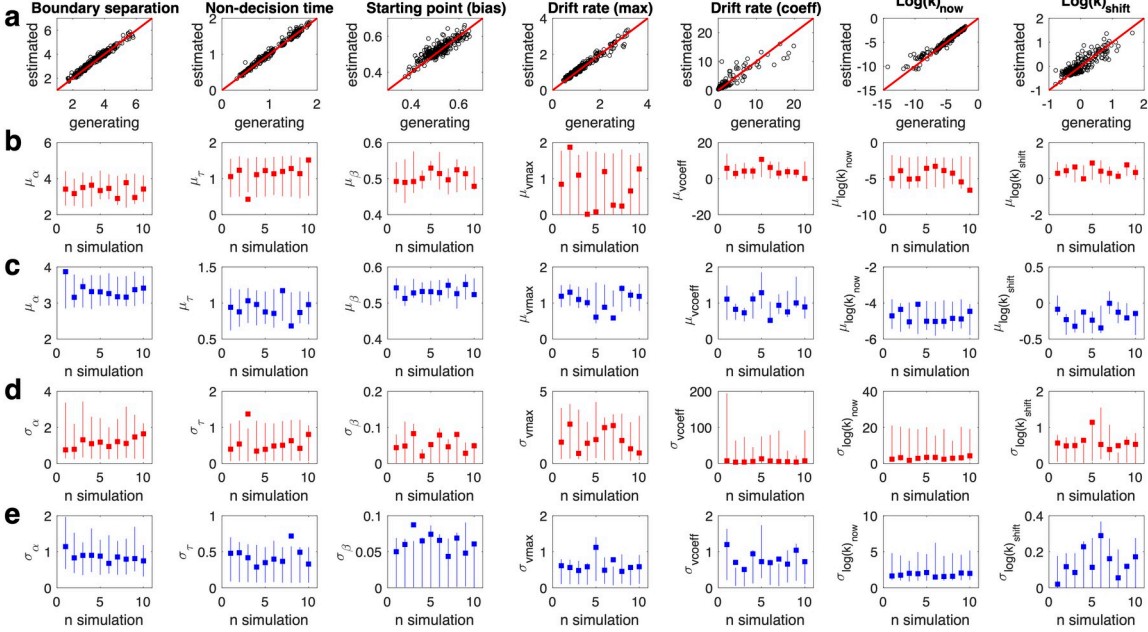

**Fig 5. Parameter recovery results for the temporal discounting DDM$_S$.** a: Recovery of subject-level model parameters pooled across all ten simulations. b/c: true generating group level means (squares) for mOFC patients (b, red) and controls (c, blue) and estimated 95% highest density intervals (lines) per simulation. d/e: generating group level standard deviations (squares) for mOFC patients (d, red) and controls (e, blue) and estimated 95% highest density intervals (lines) per simulation.

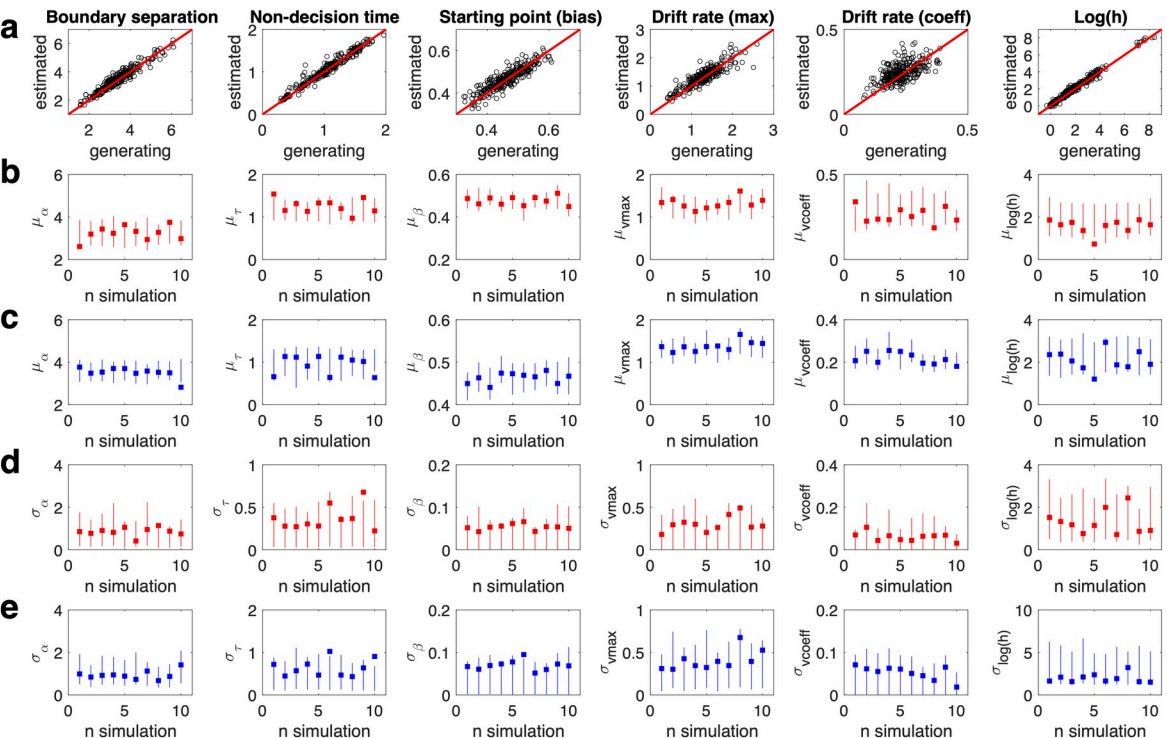

**Fig 6. Parameter recovery results for the risky choice DDM$_S$.** a: Recovery of subject-level model parameters pooled across all ten simulations. b/c: generating group level means (squares) for mOFC patients (b, red) and controls (c, blue) and estimated 95% highest density intervals (lines) per simulation. d/e: generating group level standard deviations (squares) for mOFC patients (d, red) and controls (e, blue) and estimated 95% highest density intervals (lines) per simulation.

centered at zero (Fig 7A). In contrast, for risky choice, there was evidence for a reduced boundary separation in the vmPFC/mOFC patients (Fig 8A, Table 6).

In the DDM$_S$, two components of the drift rate can be dissociated: the asymptote of the drift rate scaling function ($v_{max}$), that is, the maximum drift rate that is approached as value differences increase, and the scaling factor of the value difference ($v_{coeff}$). In both tasks, there was no evidence for a group difference in $v_{max}$ (see Table 6 and Figs 7D and 8D) and both difference distributions were centered at zero. Across tasks and groups, the value scaling parameter for the drift rate ($v_{coeff}$) was generally $> 0$, reflecting a robust positive effect of value

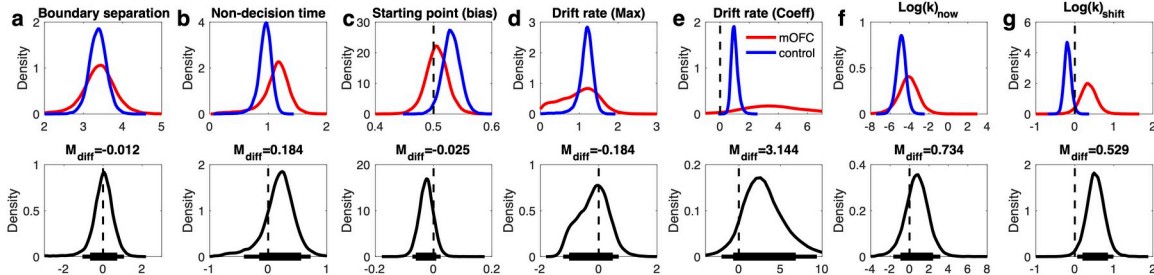

**Fig 7. Modeling results for the DDM$_S$ temporal discounting model.** Top row: posterior distributions of the parameter group means (a: boundary separation, b: non-decision time, c: starting point (bias), d: drift rate (maximum), e: drift rate (coefficient), f) log(k): discount rate in the *now* condition, g) change in log(k) in *not now* condition) for controls (blue) and mOFC patients (red). Bottom row: Posterior group differences (mOFC patients–controls) for each parameter. Solid horizontal lines indicate highest density intervals (HDI, thick lines: 85% HDI, thin lines: 95% HDI).

**Table 6. Summary of group differences in model parameters.** For each parameter and task, we report the mean difference in the group-level posterios ($M_{diff}$: patients–controls) and Bayes Factors testing for directional effects[14,46]. Bayes Factors < .33 indicate evidence for a reduction in the patient group, whereas Bayes Factors >3 indicate evidence for an increase in the patient group (see Methods section). Standardized effect sizes (Cohen's $d$) were calculated based on the posterior group-level estimates of mean and precision (see methods section).

| Model parameter | Temporal discounting | | | Risky Choice | | |
|---|---|---|---|---|---|---|
| | $M_{diff}$ | $d$ | $BF$ | $M_{diff}$ | $d$ | $BF$ |
| Boundary separation ($\alpha$) | -.012 | -.013 | 1.03 | -.368 | -.42 | .203 |
| Non decision time ($\tau$) | .184 | .44 | 4.39 | .166 | .35 | 3.52 |
| Starting point / bias ($z$) | -.025 | -.49 | .196 | .017 | .28 | 2.55 |
| Drift rate $v$ (max) | -.184 | -.26 | .647 | -.027 | -.075 | .739 |
| Drift rate $v$ (coeff) | 3.14 | 1.11 | 7.43 | .033 | .63 | 2.49 |
| Log($k$)$_{now}$ | .734 | .33 | 2.85 | - | - | - |
| Shift$_{log(k)}$ | .529 | 2.22 | 69.9 | - | - | - |
| Log($h$) | - | - | - | -.447 | -.28 | .278 |

differences on the rate of evidence accumulation (see Figs 7D and 8D). Interestingly, the drift rate scaling parameter ($v_{coeff}$) was numerically increased in the vmPFC/mOFC patients for both tasks, an effect that was substantial for temporal discounting. Here, the posterior distribution also had a higher variance compared to the control group, which was driven by 4/9 vmPFC/mOFC patients who had $v_{coeff}$ estimates that fell substantially beyond the values observed in controls and in the remaining patients (mean $v_{coeff}$ estimates: P1: 17.89, P3: 8.32, P4: 3.38, P5: 4.70). These extreme cases included the patient with the lowest discount rate (P1 $log(k)_{now}$: -10.53) and the patient with the second highest discount rate (P4 $log(k)_{now}$: -2.28).

## DDM mixture models

Both the model comparison and the posterior predictive checks suggest that choices in vmPFC/mOFC patients were still modulated by value. But the simulations showed that both very high and very low values of $v_{coeff}$ can produce RTs that are more uniform across value differences–RTs tend to be more uniformly fast for high values of $v_{coeff}$, and more uniformly slow for low values. Therefore, we additionally ran a more direct test of value sensitivity following vmPFC/mOFC damage by setting up DDM mixture models (see methods section). In short, these models allowed a proportion of trials to be produced by the $DDM_0$ and the remaining trials to be produced by the $DDM_S$, with an additional free parameter $\lambda$ controlling the mixing proportion. Notably, this analysis is agnostic with respect to the directionality of potential changes in $v_{max}$ and $v_{coeff}$, and instead solely focuses on whether groups differ in the

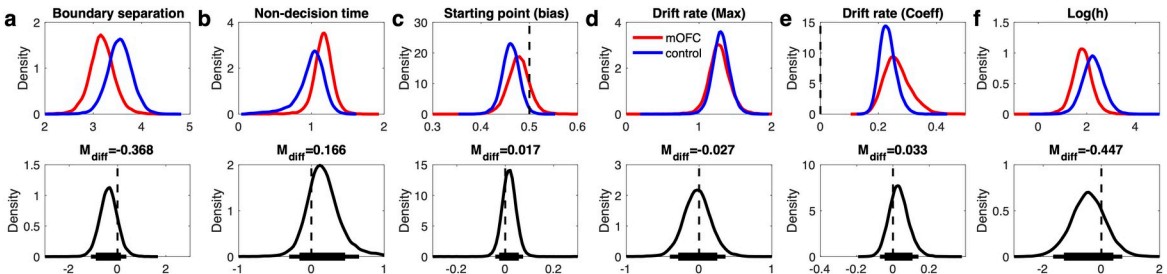

**Fig 8. Modeling results for the DDM_S risky choice model.** Top row: posterior distributions of the parameter group means (a: boundary separation, b: non-decision time, c: starting point (bias), d: drift rate (maximum), e: drift rate (coefficient), f: log(h), probability discount rate) for controls (blue) and mOFC patients (red). Bottom row: Posterior group differences (mOFC patients–controls) for each parameter. Solid horizontal lines indicate highest density intervals (HDI, thick lines: 85% HDI, thin lines: 95% HDI).

proportion of trials produced by a value-DDM vs. the $DDM_0$. Posterior distributions for $\lambda$ are shown in Fig 9. For this analysis, $\lambda$ was estimated in standard normal space and transformed to the interval [0, 1] via an inverse probit transformation on the subject level. In $z$-units, the posterior group mean of lambda was 3.67 and 4.29 in mOFC patients and controls for the temporal discounting data (Fig 9A), and 5.09 and 4.04 for the risky choice data (Fig 9B). Thus, on average, in both groups >99% of trials were better accounted for by the $DDM_S$ compared to the $DDM_0$. Because group differences in lambda are minuscule in raw proportion units, they were not further examined.

## Discussion

Here we examined different choice rules for modeling inter-temporal and risky choice / probability discounting in healthy controls and patients with vmPFC/mOFC lesions. For each task, we examined a standard softmax action selection function and three variants of the drift diffusion model (DDM). Across tasks, the data were better accounted for by a DDM with a non-linear mapping of value differences onto trial-wise drift rates ($DDM_S$) than by a DDM with linear mapping ($DDM_{lin}$) or a null model without any value modulation ($DDM_0$). Following a series of initial sanity checks (see SI), we performed detailed posterior predictive analyses, ran simulations to characterize the behavior of the $DDM_S$ in more detail and performed parameter recovery analyses. We then applied this model to reproduce our previous results on temporal discounting in patients with vmPFC/mOFC lesions[30], to characterize risk-taking behavior in these patients, and to explore group differences in DDM parameters across tasks. Finally, we examined DDM mixture models to test whether vmPFC/mOFC damage affected the proportion of trials that were best described by a value DDM as compared to the $DDM_0$.

Previous studies have successfully incorporated RTs in the modeling of value-based decision-making, e.g. via the linear ballistic accumulator model[16] or linear regression[13]. Here we build on recent work in reinforcement learning[12,14,15] and examined the degree to which the DDM could serve as the choice rule in temporal discounting and risky choice. In line with a recent model comparison in reinforcement learning[15], our model comparison of linear vs. non-linear value scaling revealed a superior fit of the DDM with non-linear (sigmoid) value scaling both for temporal discounting and risky choice data. Parameter recovery analyses showed that both subject- and group-level parameters generally recovered well. One exception were group-level variance parameters for parameters with large variability, which tended to be overestimated in some cases (though they still fell within the 95% HDIs). Posterior predictive checks of the best-fitting model revealed a good fit to the overall RT distributions of most individual participants (see S1 and S2 Figs). Given that the DDMs differed in terms of how values impact RTs, we then focused on posterior predictive checks that explicitly examined how value-dependent RTs could be reproduced by the models. While the $DDM_{lin}$ could account for some aspect of this association in some participants, in most participants the $DDM_S$ provided a superior account of the relationship between values and RTs. Specifically, the $DDM_{lin}$ in many cases overestimated RTs for smaller value differences, and underestimated RTs for very high value differences (see S6 Fig for an illustration).

One advantage of hierarchical Bayesian parameter estimation is that robust model fits can be obtained with fewer data points than are typically required for maximum likelihood estimation[47,48], and this is also the case for the drift diffusion model[47]. The reason is that in contrast to obtaining single-subject point estimates of parameters (as in maximum likelihood estimation), in hierarchical Bayesian estimation, the group-level distribution of parameters constrains and informs the parameters estimated for each individual participant. One consequence of this is shrinkage[48] or partial pooling, such that in a hierarchical model individual

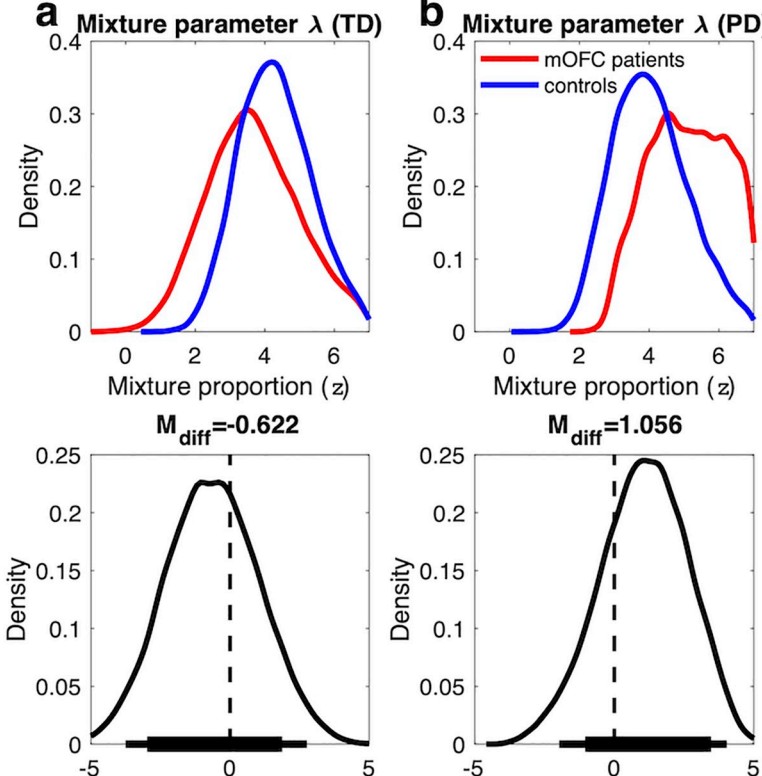

**Fig 9.** Top row: posterior distributions of the mixture parameter λ (a: temporal discounting (TD), b: risky choice / probability discounting (PD)) in z-units. Positive values of λ indicate that a greater proportion of trials was better accounted for by $DDM_S$ vs. $DDM_0$, whereas negative values indicate the reverse. λ was fitted in standard normal space with a group-level uniform prior of [−7, 7] and back-transformed on the subject-level via an inverse probit transformation. Bottom row: Posterior group differences (mOFC patients–controls) for each parameter. Solid horizontal lines indicate highest density intervals (HDI; thick lines: 85% HDI, thin lines: 95% HDI).

parameter estimates tend to be drawn towards the group mean. While this can improve the predictive accuracy of parameters, there is the possibility that meaningful between-subjects variability is removed[49]. Nonetheless, we believe that for situations with limited data per subject[47], which is a particular issue in studies involving lesion patients, the hierarchical Bayesian parameter estimation is most appropriate.

We examined variants of the DDM in tasks where they have not been applied previously (although other sequential sampling models have[16]). We therefore ran a number of initial sanity checks to validate our modeling results (see S3–S5 Figs). Additionally, analyses of the $DDM_S$ for temporal discounting reproduced our previous model-free results in vmPFC/ mOFC patients[30]: discounting behavior following vmPFC/mOFC damage was substantially more affected by SS reward immediacy than in controls, which in the present modeling scheme was reflected in a substantially increased $shift_{log(k)}$ parameter in the patient group. This reproduction of our previous results strengthens our confidence in the validity of using the DDM as the choice rule in inter-temporal and risky choice.

The temporal discounting task, but not the risky choice task, was comprised of two experimental conditions (immediate vs. delayed smaller-sooner rewards). However, we have refrained from examining condition differences in the DDM parameters in greater detail, and instead only modeled a shift parameter for log(k), rather than for the full set of DDM parameters. This was done for simplicity and in order to keep analyses comparable between tasks.

However, how contextual factors and framing effects[50,51] impact choice dynamics during inter-temporal and risky choice will be an interesting future avenue for research.

The stimulus coding scheme (coding the boundaries in terms LL/risky options vs. SS/safe options) that we adopted here differs from accuracy coding as implemented in recent applications of the DDM to reinforcement learning[14,15] (coding the boundaries in terms of correct vs. incorrect choices), with implications for the interpretation of the DDM parameters. The drift rate $v$ in the present coding scheme (as reflected in $v_{max}$ and $v_{coeff}$) can be interpreted as in classical perceptual decision-making tasks: it reflects the rate of evidence accumulation. In stimulus coding, however, higher drift rates do not directly correspond to better performance (as is the case in accuracy coding), because there is no objectively correct response. Instead the drift rate parameters reflect a participant's overall sensitivity to value differences, similar to inverse temperature parameters in softmax models. More importantly, adopting stimulus coding allowed us to estimate a starting point (bias) parameter. In all cases, the estimated bias parameters were relatively close to 0.5 (a neutral bias), but group differences for each task mirrored the results for the choice model parameters. That is, the group that displayed a preference for one option as reflected in the discount rate parameter (e.g. LL rewards in the case of controls) also exhibited a response bias towards that decision boundary. It should be noted that these numerical differences in bias could be attributable to differences in the RT distributions, differences in the binary choices, or both.

We also performed simulations to explore the impact of $DDM_S$ drift rate components on the relationship between subjective value and RTs. These simulations revealed that for very high values of $v_{coeff}$ the $DDM_S$ produces longer RTs only for then highest conflict choices (green lines in Fig 4, this effect can also be seen in P1 in Fig 2, the participant with the highest $v_{coeff}$ for temporal discounting of all participants). In contrast very low values of $v_{coeff}$ yield RTs that tend to be uniformly longer for all but the easiest (highest value-difference) choices. The implication is that increases and decreases in $v_{coeff}$ cannot unambiguously be interpreted as increases and decreases in value-sensitivity in RTs. Rather, as the simulations show, value-sensitivity (if interpreted as the degree of RT deceleration with increasing conflict) is maximal for intermediate values of $v_{coeff}$. At the same time, the magnitude of this effect depends on $v_{max}$.

Our results provide novel insights into the role of the vmPFC/mOFC in value-based decision-making. Our DDM analyses show a comparable maximum drift rate $v_{max}$ in the two groups for both tasks, while $v_{coeff}$ was increased in the patients for temporal discounting. However, examination of posterior predictive checks for each individual lesion patient (Figs 2 and 3) shows that RTs were modulated by value in most patients, and that this modulation was better accounted for the $DDM_S$ than $DDM_{lin}$. This suggests that value sensitivity of RTs was intact in the patients. This interpretation is corroborated by the DDM mixture model analyses: in both groups, the vast majority of trials was better accounted for by the $DDM_S$ than the $DDM_0$, with no evidence for a group difference in these mixture proportions. This is in line with an earlier report showing reduced preference consistency but no changes in overall RTs or the value-modulation of RTs in vmPFC/mOFC patients[40]. If one considers the overwhelming evidence of neuroimaging studies showing a prominent role of the vmPFC/mOFC in reward valuation[42,43], it is nonetheless striking that lesions to this region do not negatively impact the value-sensitivity of the evidence accumulation process during value-based decision-making. Our data are therefore more compatible with the idea that vmPFC/mOFC, likely in interaction with other regions[52,53], plays a role in self-control, such that lesions shift preferences towards options with a greater short-term appeal.

Previous work has suggested that damage to vmPFC/mOFC might decrease the temporal stability of value representations, leading to inconsistent preferences[39–41]. There was no evidence in the present data that the lesion patients' decisions were more "noisy" or "erratic".

Similar to a previous study on temporal discounting[31], choice consistency was high such that the best-fitting $DDM_S$ accounted for about 90% of binary choices in both groups and tasks, suggesting that value representations on a given trial[40] and throughout the course of the testing sessions were relatively stable in both groups. In contrast, results from both tasks revealed an increase in non-decision times in the patient group. Whether this effect is specific to value-based decisions or extends to other choice settings is an open question. However, accounts of perceptual decision-making have typically focused on lateral prefrontal cortex regions[54,55]. Together, these observations suggest that vmPFC/mOFC lesions lead to a slowing of more basic perceptual and/or response-related processes during value-based decision-making, while leaving the effects of value-differences on the evidence accumulation process strikingly intact.

Previous studies have shown increases in risky decision-making following vmPFC/mOFC damage[33,35]. Our finding of attenuated discounting over probabilities in the patients is consistent with these previous results. However, our model-based analysis revealed an additional effect: lesion patients also exhibited reduced response caution during risky choice, reflected in a reduced boundary separation parameter. In contrast, this was not observed for temporal discounting. This suggests that risk taking in vmPFC/mOFC patients might not only be driven by altered preferences, but also by more premature responding.

Taken together, our results demonstrate the feasibility of using the DDM as the choice rule in the context of inter-temporal and risky decision-making. Model comparison revealed that a variant of the DDM that included a non-linear drift rate modulation provided the best fit to the data. We further show that the application of a sequential sampling model revealed additional insights: while the value-dependency of the evidence accumulation process was strikingly unaffected by vmPFC/mOFC damage, we observed a slowing of non-decision times both in temporal discounting and risky choice, with implications for models of decision-making. This modeling framework might provide further insights, e.g. when studying mechanisms underlying context-dependent changes in decision-making[50,56–58] or impairments in decision-making in psychiatric[59][59] and neurological disorders[6].

## Materials & methods

### Ethics statement

All participants gave informed written consent, and the study procedure was approved by the local institutional review board of the University of California, Berkeley, USA.

### Procedure

We report data from two value-based decision-making tasks: one previously unpublished data set from a risky-choice task and one previously published data set from a temporal discounting task (see below for task details). Data were acquired in nine patients with focal lesions that included medial orbitofrontal cortex and nineteen healthy age- and education-matched controls. The temporal discounting task was always performed first, followed by the risky choice task.

For a detailed account of etiology, socio-demographic information for all participants and lesion location data for the patients, the reader is referred to our previous paper[30].

### Temporal discounting task

Here participants performed 224 trials of an inter-temporal choice task involving a series of choices between smaller-but-sooner (SS) and larger-but-later (LL) rewards. On half the trials,

the SS reward was available immediately (*now* condition), whereas on the other half of the trials, the SS reward was available only after a 30d delay (*not now* condition). In the *now* condition, the SS reward consisted of $10 available immediately and LL rewards consisted of all combinations of fourteen reward amounts (10.1, 10.2, 10.5, 11, 12, 15, 18, 20, 30, 40, 70, 100, 130, 150 dollars) and seven delays (1, 3, 5, 8, 14, 30, 60 days). Trials for the *not now* condition where identical, with the exception that an additional delay of 30 days was added to both options, such that in *not now* trials, the SS reward was always associated with a 30 day delay, and LL reward delays ranged from 31 to 91 days. Trials were presented in randomized order and with a randomized assignment of options to the left/right side of the screen. Options remained on the screen until a response was logged.

## Risky choice task

Here participants made a total of 112 choices between a certain (100% probability) $10 reward and larger-but-riskier options. The risky options consisted of all combinations of sixteen reward amounts (10.1, 10.2, 10.5, 11, 12, 15, 18, 20, 25, 30, 40, 50, 70, 100, 130, 150 dollars) and seven probabilities (10%, 17%, 28%, 54%, 84%, 96%, 99%). Trials were presented in randomized order and with a randomized assignment of options to the left/right side of the screen. As in the temporal discounting task, options remained on the screen until a response was logged.

Participants were instructed that all choices from the two tasks were potentially behaviorally relevant. A single trial was pseudo-randomly selected following completion of both tasks, and participants received their choice from that trial as a cash bonus.

## Temporal discounting model

Based on previous work on the effect of smaller-sooner (SS) reward immediacy on discounting behavior [60,61], we hypothesized discounting to be hyperbolic relative to the soonest available reward. Previous studies[30,61] fitted separate discount rate parameters to trials with immediate vs. delayed SS rewards. Here we extended this approach by instead fitting a single k-parameter (reflecting discounting in the *now* condition), and a subject-specific shift parameter *s* modeling the reduction in log(k) in the *not now* condition as compared to the *now* condition:

$$SV(LL)_t = \frac{A_t}{(1 + (\exp(k - I_t * s)) * IRI_t)} \tag{1}$$

Here, *SV* is the subjective discounted value of the delayed rewards, *A* is the amount of the *LL* reward on trial t, *k* is the subject specific discount rate for *now* trials in log-space, *I* is an indicator variable coding the condition (0 for *now* trials, 1 for *not now* trials), *s* is a subject-specific shift in log(*k*) between *now* and *not-now* conditions and *IRI* is the inter-reward-interval on trial *t*. Note that this model corresponds to the elimination-by-aspects model of Green et al. [60].

## Risky choice model

Here we applied a simple one-parameter probability discounting model[62,63], where discounting is hyperbolic over the odds-against-winning the gamble:

$$SV(risky_t) = \frac{A_t}{1 + \exp(h) * \theta_t}, with\ \theta_t = \frac{1 - p_t}{p_t} \tag{2}$$

Here SV is the subjective discounted value of the risky reward, *A* is the reward amount on trial *t* and *θ* is the odds-against winning the gamble. The probability discount rate *h* (again

fitted in log-space) models the degree of value discounting over probabilities. We also fit the data with a two-parameter model that includes separate parameters for the curvature and elevation of the probability weighting function[64–66]. However, when fitting a two-parameter model at the single subject level, in a number of individual cases the posterior distributions of the curvature and/or elevation parameters were not clearly peaked, suggesting that we likely did not have adequate coverage of the probability and amount dimensions to reliably dissociate these different components of risk preferences. For this reason, we opted for the simpler single-parameter hyperbolic model instead.

## Softmax choice rule

Standard softmax action selection models the probability of choosing the LL reward (or the risky option) on trial $t$ as:

$$P(LL)_t = \frac{e^{\beta * SV(LL_t)}}{e^{\beta * SV(LL_t)} + e^{\beta * SV(SS_t)}} \tag{3}$$

Here, $SV$ is the subjective value of the LL reward according to Eq 1 (or the risky reward according to Eq 2) and $\beta$ is an inverse temperature parameter, modeling choice stochasticity (for $\beta = 0$, choices are random and as $\beta$ increases, choices become more dependent on the option values).

## Drift diffusion choice rule

For the DDMs, we build on earlier work in reinforcement learning[14,15] and inter-temporal choice[13,16]. Specifically, we replaced the softmax action selection rule (see previous section) with the DDM as the choice rule, using the Wiener module[67] for the JAGS software package [68]. In contrast to previous reinforcement learning approaches[14,15] that used accuracy coding for the boundary definitions, we here used stimulus coding, such that the lower boundary was defined as a selection of the SS reward (or the 100% option in the case of risky choice), and the upper boundary as selection of the LL reward (or the risky option in the case of risky choice). This is because we were explicitly interested in modeling a bias towards SS vs. LL options. RTs for choices towards the lower boundary were multiplied by -1 prior to estimation.

We initially used absolute RT cut-offs for trial exclusion[14] such that 0.4s < RT < 10s. However, when using such an absolute cut-off, single fast outlier trials can still force the non-decision-time to adjust to accommodate these observations, which can lead to a massive negative impact on model fit at the individual-subject level. This is also what we observed in two participants when plotting posterior predictive checks from hierarchical models with absolute cut-offs. For this reason, we instead excluded for each participant the slowest and fastest 2.5% of trials from analysis, which eliminated the problem. The RT on trial $t$ is then distributed according to the Wiener first passage time (*wfpt*):

$$RT_t \sim wfpt(\alpha, \tau, z, v) \tag{4}$$

Here, $\alpha$ is the boundary separation (modeling response caution / the speed-accuracy trade-off), $z$ is the starting point of the diffusion process (modeling a bias towards one of the decision boundaries), $\tau$ is the non-decision time (reflecting perceptual and/or response preparation processes unrelated to the evidence accumulation process) and $v$ is the drift rate (reflecting the rate of evidence accumulation). Note that in the JAGS implementation of the Wiener model [67], the starting point $z$ is coded in relative terms and takes on values between 0 and 1. That

is, $z = .5$ reflects no bias, $z > .5$ reflects a bias towards the upper boundary, and $z < .5$ a bias towards the lower boundary.

In a first step, we fit a null model (DDM$_0$) that included no value modulation. That is, the null model for both the temporal discounting and risky choice data had four free parameters ($\alpha, \tau, v$, and $z$) that for each participant were constant across trials.

Next, to link the diffusion process to the valuation models (Eq 1, Eq 2), we compared two previously proposed functions linking trial-by-trial variability in value differences to the drift rate. First, we used a linear mapping as proposed by Pedersen et al. (2017)[14]:

$$v_t = v_{coeff} * (SV(LL_t) - SV(SS_t)) \tag{5}$$

Here, $v_{coeff}$ is a free parameter that maps value differences onto the drift rate $v$ and simultaneously transforms value differences to the appropriate scale of the DDM[14]. This implementation naturally gives rise to the effect that highest conflict (when values are highly similar) would be expected to be associated with a drift rate close to zero. For positive values of $v_{coeff}$, as SV(SS) increases over SV(LL), the drift rate becomes more negative, reflecting evidence accumulation towards the lower (SS) boundary. The reverse is the case as SV(LL) increases over SV(SS). For the risky choice models, SV(LL) was replaced with SV(risky), and SV(SS) with SV(safe). Second, we also applied an additional non-linear transformation of the scaled value differences via the S-shaped function suggested by Fontanesi et al. (2019) [15]:

$$v_t = S(v_{coeff} * (SV(LL_t) - SV(SS_t))) \tag{6}$$

$$S(m) = \frac{2 * v_{max}}{1 + e^{-m}} - v_{max} \tag{7}$$

S is a sigmoid function centered at 0 with $m$ being the scaled value difference from Eq 6, and asymptote $\pm v_{max}$. Again, effects of choice difficulty on the drift rate naturally arise: for highest decision conflict when SV(SS) = SV(LL), the drift rate would again be zero, whereas for larger value differences, $v$ increases up to a maximum of $\pm v_{max}$. Table 7 provides an overview of the parameters of the DDM$_S$ model.

## DDM mixture models

As a further test of whether groups differed with respect to the degree to which RT distributions showed value sensitivity, we also examined mixture models to explore whether the proportion of trials best accounted for by the best-fitting value DDM (DDM$_S$) vs. the null model (DDM$_0$) differed between groups. Mixture models contained the full hierarchical parameter sets of both the DDM$_S$ and DDM$_0$, as well as a mixture parameter $\lambda$, such that a proportion of $\lambda$ trials were allowed to be accounted for by the DDM$_S$ and 1-$\lambda$ trials by the DDM$_0$. For each group, the prior mean for $\lambda$ was set to a uniform distribution [–7, 7] and subject level parameters were drawn from a normal distribution and transformed via an inverse probit transformation to the interval [0, 1].

## Hierarchical Bayesian models

We used the following model-building procedure. In a first step, models were fit at the single-subject level. After validating that reasonably good fits could be obtained for single-subject data (by ensuring that $\hat{R}$ statistic was in an acceptable range of $1 \leq \hat{R} \leq 1.01$ and the posterior distributions were centered at reasonable parameter values) we re-fit all models using a hierarchical framework with separate group-level distributions for controls and patients. We again

**Table 7. Overview of the parameters of the DDM$_S$ models and priors for group means.**

| Parameter | DDM$_S$: Temporal discounting | DDM$_S$: Risk taking | Group-level prior (μ) |
|---|---|---|---|
| α | \multicolumn Boundary separation | | Uniform (.01, 5) |
| τ | Non-decision-time | | Uniform (.1, 6) |
| z | Bias (>.5: LL, < .5: SS) | Bias (>.5: risky, < .5: safe) | Uniform (.1, .9) |
| $v_{coeff}$ | Drift rate: value-difference scaling | | Uniform (-100, 100) |
| $v_{max}$ | Drift rate: maximum | | Uniform (0,100) |
| log(k) | Discount rate now-trials | - | Uniform (-20,3) |
| $shift_{log(k)}$ | log(k) reduction not-now | - | Gaussian (0, 2) |
| log(h) | - | Probability discount rate | Uniform (-10, 10) |
| λ | Mixture parameter (proportion of DDM$_S$ trials) | | Uniform (-7,7) |

assessed chain convergence such that values of $1 \leq \hat{R} \leq 1.01$ were considered acceptable for all group- and individual-level parameters. As priors for the group-level hyperparameters we used uniform distributions for means defined over numerically plausible ranges (see Table 7) and gamma distributions with shape and rate parameters .001 for precision. Individual-subject parameters were then drawn from normal distributions with group-level means and precision.

## Model estimation and comparison

All models were fit using Markov Chain Monte Carlo (MCMC) as implemented in JAGS[68] with the *matjags* interface (https://github.com/msteyvers/matjags) for Matlab (The Mathworks) and the JAGS Wiener package[67]. For each model, we ran two chains with a burn-in period of 50k samples and thinning of 2. 10k further samples were then retained for analysis. Chain convergence was assessed via the $\hat{R}$ statistic, where we considered $1 \leq \hat{R} \leq 1.01$ as acceptable values. Relative model comparison was performed using the *loo* R package[44], and we report both WAIC and the estimated log pointwise predictive density (elpd) which estimates the leave-one-out cross-validation predictive accuracy of the model[44].

## Posterior predictive checks

Because a superior relative model fit does not necessarily mean that the best-fitting model captures key aspects of the data, we additionally performed posterior predictive checks. To this end, during model estimation, we simulated 10k full datasets from the hierarchical models, based on the posterior distribution of the parameters. We then compared these simulated data to the observed data in two ways. First, to visualize how models accounted for the overall observed RT distributions, a random sample of 1k of the simulated data sets were smoothed via non-parametric density estimation in Matlab (*ksdensity.m*) and overlaid on the observed RT distributions for each individual participant. Second, we examined how the different DDM models accounted for the observed association between RT and value. To this end, we binned trials into five bins based on the subjective value of the larger-later or risky reward (as per Eqs 1 and 2) for each individual participant, and for these bins again compared observed mean RTs to model-predicted RTs from the simulations.

## Parameter recovery analyses

For models of decision-making, identifiability of the true data generating parameters is a crucial issue [48]. We therefore conducted parameter recovery simulations for the most complex model, the DDM$_S$. We selected ten random datasets simulated from the posterior distributions, and re-fit these datasets with the generating model using the same methods as outlined

above. The recovery of subject-level parameters was examined by plotting generating parameters against estimated parameters. The recovery of group-level parameters was examined overlaying the true generating group-level means over the 95% highest-density intervals of the posterior distributions.

### Simulating effects of drift rate components on RTs

To gain additional insights into how drift rate components $v_{max}$ and $v_{coeff}$ of the DDM$_S$ affect RT distributions and the value-dependency of RTs more specifically, we ran additional simulations. Specifically, we simulated 50 RTs from the DDM$_S$ for each of 400 value differences ranging from zero to ± 20. We ran 30 simulations in total, systematically varying $v_{max}$ and $v_{coeff}$ while keeping the other DDM parameters (boundary separation, bias, non-decision time) fixed at mean posterior values of the control group (see Table 6). For each simulated data set, we examined the shape of the overall RT distribution, the degree to which RTs depended on value differences, and the proportion of binary choices (lower vs. upper boundary) as a function of value differences.

### Analysis of group differences

To characterize group differences, we show posterior distributions for all parameters, as well as 85% and 95% highest density intervals for the difference distributions of the group posteriors. We furthermore report Bayes Factors for directional effects[14,46] based on these difference distributions as $BF = i/(1−i)$ were $i$ is the integral of the posterior distribution from 0 to $+\infty$, which we estimated via non-parametric kernel density estimation in Matlab (*ksdensity. m*). Following common criteria[69], Bayes Factors > 3 are considered positive evidence, and Bayes Factors > 12 are considered strong evidence. Bayes Factors < 0.33 are likewise interpreted as evidence in favor of the alternative model. Finally, we report standardized measures of effect size (Cohen's *d*) calculated based on the mean posterior distributions of the group means and the pooled standard deviations across groups.

### Code availability

JAGS model code for all models is available on the Open Science Framework (https://osf.io/5rwcu/).

### Supporting information

**S1 Fig. Posterior predictive plots of the drift diffusion temporal discounting model with non-linear value scaling of the drift rate (DDM$_S$) for all participants (red–mOFC patients, blue–controls).** Histograms depict the observed RT distributions for each participant. The solid lines are smoothed histograms of the model predicted RT distributions from 1000 individual subject data sets simulated from the posterior distribution of the best-fitting hierarchical model. RTs for smaller-sooner choices are plotted as negative, whereas RTs for larger-later choices are plotted as positive. The x-axes are adjusted to cover the range of observed RTs for each participant.
(TIF)

**S2 Fig. Posterior predictive plots of the drift diffusion probability discounting / risky choice model with non-linear value scaling of the drift rate (DDM$_S$) for all participants (red–mOFC patients, blue–controls).** Histograms depict the observed RT distributions for each participant. The solid lines are smoothed histograms of the model predicted RT distributions from 1000 individual subject data sets simulated from the posterior distribution of the

best-fitting hierarchical model. RTs for choices of the safe option are plotted as negative, whereas RTs for risky choices are plotted as positive. The x-axes are adjusted to cover the range of observed RTs for each participant.
(TIF)

**S3 Fig. Consistency of model parameters for temporal discounting (TD: a/b) and probability discounting (PD, c) between softmax and DDM$_S$ choice rules.** Scatter plots (controls: blue, mOFC patients: red) show model parameters estimated via a standard softmax choice rule (x-axis) vs. parameters estimated via a drift diffusion model choice rule with non-linear drift rate scaling (DDM$_S$, y-axis). a) Temporal discounting log(discount rate) for *now* trials. b) Shift in log(k) between *now* and *not now* trials). c) Probability discounting log(discount rate).
(TIF)

**S4 Fig. Associations between model-based non-decision time and model-free response times.** Scatter plots (red mOFC patients, blue: controls) depict associations between model-based non-decision time from the best fitting DDM$_S$ models (x-axis) and minimum RT (a/b) and median RT (c/d) for temporal discounting (a/c) and risky choice / proability discounting (b/d).
(TIF)

**S5 Fig. Associations between model-based boundary separations and model-free response times.** Scatter plots (red: mOFC patients, blue: controls) depict associations between model-based boundary separation from the best fitting DDM$_S$ models (x-axis) and minimum RT (a/b) and median RT (c/d) for temporal discounting (a/c) and risky choice / proability discounting (b/d).
(TIF)

**S6 Fig. Illustration of the differential effects of linear vs. sigmoid drift rate scaling.** Linear scaling predicts longer RTs (lower drift rates) than sigmoid scaling for all but the greatest value differences, where the effect reverses. The reversal point depends on the drift rate components (DDM$_S$1: $v_{max}$ = 1.1786, $v_{coeff}$ = .997, DDM$_S$2: $v_{max}$ = .6, $v_{coeff}$ = .2). The dashed line marks a value difference of -10, which was the lower bound of value differences in the present experimental design (i.e., the case when the risky or larger-later option was discounted to almost 0).
(TIF)

**S7 Fig.** Associations between drift rate components and discount rates for temporal discounting (a) and risky choice / probability discounting (b). Top panels show $v_{max}$ and lower panels show $v_{coeff}$.
(TIF)

**S8 Fig.** Simulated temporal discounting response time distributions (left) and mean predicted response times per value bin (right) for a virtual participant for different values of $v_{max}$ and $v_{coeff}$. See S1 Table (left column) for parameter values.
(TIF)

**S9 Fig.** Simulated risky choice response time distributions (left) and mean predicted response times per value bin (right) for a virtual participant for different values of $v_{max}$ and $v_{coeff}$. See S1 Table (right column) for parameter values.
(TIF)

**S1 Table. Parameter values used for simulation analyses depicted in S8 and S9 Figs.** All parameters are the posterior group means of the control group, with the exception of log(k)$_{now}$

and the two drift rate modulator variables, which were selected for illustrative purposes.
(DOCX)

**S1 Text. Model validation analyses: associations of DDM parameters with model-free measures.**
(DOCX)

**S2 Text. Associations between drift rate components and discount rates.**
(DOCX)

## Acknowledgments

We thank Donatella Scabini for help with patient recruitment, Natasha Young for help with testing control subjects and all members of the Peters Lab at University of Cologne for helpful discussions.

## Author Contributions

**Conceptualization:** Jan Peters, Mark D'Esposito.

**Data curation:** Jan Peters.

**Formal analysis:** Jan Peters.

**Funding acquisition:** Jan Peters.

**Investigation:** Jan Peters.

**Methodology:** Jan Peters.

**Project administration:** Jan Peters.

**Writing – original draft:** Jan Peters.

**Writing – review & editing:** Jan Peters, Mark D'Esposito.

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
