## [Decision Letter · Decision Letter 0]

2 Aug 2019

Dear Dr Peters,

Thank you very much for submitting your manuscript 'The drift diffusion model as the choice rule in inter-temporal and risky choice: a case study in medial orbitofrontal cortex lesion patients and controls.' for review by PLOS Computational Biology. Your manuscript has been fully evaluated by the PLOS Computational Biology editorial team and in this case also by independent peer reviewers. The reviewers appreciated the attention to an important problem, but raised some substantial concerns about the manuscript as it currently stands. While your manuscript cannot be accepted in its present form, we are willing to consider a revised version in which the issues raised by the reviewers have been adequately addressed. We cannot, of course, promise publication at that time.

Note that PLoS will usually require data to be made available freely, if at all possible. Is it possible to anonymise the data, so that it can be made freely available?

Sincerely,

Ulrik R. Beierholm

Associate Editor

PLOS Computational Biology

Samuel Gershman

Deputy Editor

PLOS Computational Biology

[LINK]

Reviewer's Responses to Questions

**Comments to the Authors:**

Reviewer #1: In the manuscript entitled “The drift diffusion model as the choice rule in inter-temporal and risky choice: a case study in medial orbitofrontal cortex lesion patients and controls”, Peters & D’Esposito present some novel insights by fitting cognitive models to partly unpublished data. These insights include: (1) vmPFC/mOFC damage patients have longer non-decision times and reduced decision caution compared to healthy and age- and education-matched controls; (2) they show increased risk-taking;

(3) the best way to account for value-based decisions in an inter-temporal choice task as well as in a risk-taking task is to modify the DDM so that the rate of evidence accumulation is proportional to the difference in subjective value between the two options; (4) the mapping between value-differences and the rate of evidence accumulation is non-linear rather than linear.

The present manuscript offers new insights in addition to the current literature in value-based decision making. However, there are a number of points (major and minor ones) that I think should be addressed by the authors.

Major points:

To test the hypothesis that the “vmPFC/mOFC damage might also render RTs during decision-making less dependent on value” it would be better to use a mixed-modelling approach rather than simply looking for differences in the drift-value-coefficient between groups of participants. This is because a low vs. high drift-value-coefficient means that decisions are still based on values, but are less sensitive to value differences. What would be interesting to see, is whether for a higher portion of trials compared to control subjects participants with vmPFC/mOFC damage can be simply described by the null-DDM instead of by the value-DDM. This could be tested with a mixed model, where a parameter (e.g., lambda) could control the proportion of trials that can be best described by a null-DDM vs. a value-DDM. This would allow to formulate and test the hypothesis that the lambda would be higher in participants with a vmPFC/mOFC damage.

It seems to me that the null model (DDM0) is too simple, making the comparison with the two value-based DDMs unfair. I strongly suggest to fit separate sets of the 4 parameters (alpha, tau, v, and z) across conditions (Now vs. Not now) for the inter temporal choice task. Since there are no clear conditions in the risky choice task, I am not sure how this could be done there.

Regarding the prior distributions, I have 2 suggestions: (1) that the priors are properly written and described in the text or appendix, or in the supplementary materials. They should be easily accessible by a reader and should not just be retrievable within the online code. Note that it is also necessary to specify the priors for the individual parameters, depending on the group parameters; (2) uniform priors are not uninformative, especially when restricted to sensible ranges. This suggests that the authors had an idea of which values were sensible and could therefore specify weakly-informative priors (see Gelman et al., 2014). I suggest to use “sensibly” centered Cauchy distributions – since they have the advantage of being heavy-tailed distributions and therefore weakly-informative – for the means and Half-Cauchy for the standard deviations.

The authors should add a parameter recovery section for the winning model (as in, e.g., Pedersen et al., 2017, Fontanesi et al., 2019a, and Fontanesi er al., 2019b). This is quite crucial when proposing new, complex models such as value based modifications of the DDM.

The posterior predictive checks are hard to assess. To better assess them, the authors should group mean RTs or RT quantiles for SS vs. LL options or for risky vs. safe options by condition and compare them with the same summary statistics on the data (as in, e.g., Fontanesi et al., 2019a, Fontanesi et al., 2019b)

For the Cohen’s d calculation, the authors should use the whole posterior traces for mean and pooled standard deviation, instead of the mean of the posterior distributions. This would make the calculation “more Bayesian” and more reliable.

I would exclude from the analyses correlations between DDM parameter estimates and model-free RT statistics. The DDM is supposed to decompose RT and choices distributions into interpretable parameters so: (1) some parameters always correlate to some extent to RTs (2) it is unclear to me what this would add to the interpretation of the model. If the aim was to check for qualitative model fit, the posterior predictive checks should be enough.

Minor points:

Please substitute every recurrence of “reaction time” with “response time”, which is more appropriate in this case (as this is not a mere stimulus reaction task but participants need to integrate information in order to make a decision). In general, I would only write response time (RT) at the first recurrence, and only refer to it as RT or RTs after that.

Regarding model comparison, I would suggest to switch to WAIC or LOO, which also allow you to have an estimate of the error in such measures and more reliably assess the difference in model fit between two models (as in, e.g., Pedersen et al., 2017 and Fontanesi et al., 2019a). This can be easily achieved via the R loo package, that only needs the mcmc traces as input (you could save your analyses output from Matlab and load them in R…).

I do not understand what it means that the distributions should have a clear peak or a clearly Gaussian shape: not all parameter distributions are supposed to have such a shape, so this shouldn’t be a way to asses model recoverability. On the contrary, looking at chain convergence and parameter recovery are.

The “m” in Equation 7 is not the slope of the sigmoid function, as described in the main text, but is the input value-difference that is transformed by the sigmoid function. The slope would then be Vcoeff (Vmod in Fontanesi et al., 2019a).

R-hats statistics should just be between 1 and some value close to 1. So please correct to 1 <= Rhat <= 1.01

In the results, clarify this sentence: “Since the correlation for shiftlog(k) appeared to be somewhat inflated by the extreme datapoints of the mOFC patients, we re-ran the correlation only in the control group. Here, the correlation was lower but still robust (r=.52).” What are these extreme datapoints? What was the correlation in the mOFC patients?

Very minor points:

In the abstract, refer to “Bayesian parameter estimation” instead of “Bayesian estimation scheme”.

In the introduction, remove the word “usually” in the first sentence of the second paragraph. DDM can only have 2 response boundaries, otherwise it would be a different model. In the last sentence of the same paragraph substitute “simple” with 2-alternatives forced choice tasks (this are what the DDM was made for).

I think it’s a bit misleading to call the DDM boundaries 0 and 1, so if possible I would delete that, or better explain that the lower boundary corresponds to when the accumulated evidence is equal to 0, the upper boundary is who the accumulated evidence is equal to alpha and the starting point z is half alpha.

Put a comma after every recurrence of e.g. (e.g., )

References:

Fontanesi, L., Gluth, S., Spektor, M.S. et al. Psychon Bull Rev (2019a). https://doi.org/10.3758/s13423-018-1554-2

Fontanesi, L., Palminteri, S. & Lebreton, M. Cogn Affect Behav Neurosci (2019b) 19: 490. https://doi.org/10.3758/s13415-019-00723-1

Gelman, A., Carlin, J. B., Stern, H. S., & Rubin, D. B. (2014) Bayesian data analysis, (3rd edn.) London: Chapman & Hall/ CRC.

Pedersen, M. L., Frank, M. J., & Biele, G. (2017). The drift diffusion model as the choice rule in reinforcement learning. Psychonomic Bulletin & Review, 24(4), 1234—1251.

Reviewer #2: In the current study, the authors are using drift-diffusion modeling to predict a combination of choice and RT data, from two reinforcement learning tasks (temporal/probability discounting) performed by healthy controls and mOFC/vmPFC lesion patients. The authors suggest that DDM can be adequately used to describe observed data, and that it does not fall behind compared to a more conventional RL model, describing only choice behavior. The authors then suggest that (a) a ddm with a non-linear mapping between subjective-value and drift-rate provide the best fit to the data compared to more conventional ddm models. (b) Group differs mainly in increased non-decision time, and reduced decision threshold for patients vs. controls. The authors report no group differences in value processing.

I believe this is a valuable study, both in the sense that it aims to contribute by examining the applicability of evidence accumulation modeling to described choice&RT data, and presents interesting results with lesion patients. However, I think there are some issues that needs to be further elaborated and explored:

1. The authors conclude that non-linear drift rate modulation provided the best fit to the data. This is a great finding, but I think that it is very important to understand why that is, in the mechanistic level, specifically in terms of the relationship between subjective value and decision-time:

a. At the moment, it is hard to figure out why the non-linear aspect of the DDMs allows a better description for the data. Is this because the RT association with value differences between the two options is stronger for lower value differences? Or maybe this is due to the fact that some participants are less sensitive to the value manipulation (i.e., resulting in a very high Vcoef)?

b. Does the better fit for DDMs vs. DDMlin comes only from choice data, or does it provide better fit to RT data as well? I believe this is important to fully understand what part of the observed data is better explained by DDMs (e.g., it might not be about RTs at all, with DDMs better accounting for choice data alone compared to DDMlin). If it is mostly due to choice data, I am not sure why the use of ddm is justified here.

c. On the same point above, does vmax/vcoef estimates reflected differently in different aspects of the RT distribution (e.g., the tail/leading edge of the distribution)? Maybe the authors can also include a simulation where vmax/vcoef are mapped to aspects of the RT distribution (e.g., using ex-Gaussian fitting). This might then be used to show why DDMs actually fit better with RT data compared to DDMlin (hoping it does fit better to both, rather than to choice behavior only).

d. What is the relationship between vcoef/vmax and discounting parameters? Does DDMs provide a better fit because it help to better capture the RT-SV associations – or rather help account for individuals where such a relationship is actually absent (e.g., maybe by allowing a high vcoef?)

2. The authors suggest two analysis which led them to the conclusion that DDM can be adequately used to describe value based decision (depicted in Fig4 and Fig5/6). However, I feel that these analyses are more of a 'sanity checks' rather than novel results:

a. The authors report high correlations between the same parameters fitted with either softmax or DDMs. Yet, since both models describe choice behavior – why would we expect the same parameter to differ due to the modeling of choice and RT combination vs choice only? I think this needs better justification/explanation. What did you have in mind? Did you expect RTs to change the model ability to accurately predict choice for some reason? I think the challenge here is not to show that DDM account for choices similar to softmax models (which means that modeling a combination of choice and RT as opposed to choice only, doesn't reduce the fit for choice data). The challenge here in my mind, is to show the benefits of modeling RTs and choices at the same time. Since modeling both RT and choice is more difficult, I think it should be justified by laying down the possible advantages of using such an approach.

b. Fig 5/6. Why would we expect anything else then a positive correlation between nd /th and min/median/(mean?) RT? I appreciate the fact that the authors are including this – but I don't see how this is more than a sanity check. Did you have any reason to believe that a value based DDM model will tamper with the nd/th relationship with RT? Why?

c. Further re "we re-ran the correlation only in the control group. Here, the correlation was lower but still robust (r=.52)." I would suggest this is very low. Why is that? Maybe it has to do with the recoverability of this parameter specifically, or maybe very low effect/variance between conditions for controls? The point is that this might be unrelated to the difference between DDMs and softmax models.

3. Group differences look very interesting and valuable, but it's not clear whether you use these to validate the use of DDM, justifying the use of DDM for value based-decisions, or add to the mOFC literature per-se:

a. If group differences in nd is unrelated to value based processes – why is this a demonstration of why DDM can be beneficial here? Couldn't this be done with perceptual based decisions as well?

b. The authors report similar vmax, but higher vcoef for mOFC group. They conclude that "value-differences exert a similar (if not stronger) effect on trial-wise drift rates in vmPFC/mOFC patients compared to controls". I am not sure I understand why this led them to conclude that value choice processes are intact in mOFC patients. I think high vcoef might actually be a way of the model to account for participants that are insensitive to value differences.

c. Is it possible that the difference in starting point is only due to choice data, but not RT?

4. The issue of accuracy vs. stimulus coding is emphasized. However, it was hard for me to follow why this makes a difference. The assignment of the upper boundary to the higher value option is, to the best of my understating, strictly technical (both models are perfectly equivalent). I think it's good to note the differences, but I'm not sure I understand why it emphasized (i.e., had a paragraph both in the intro and discussion).

**Have all data underlying the figures and results presented in the manuscript been provided?**

Reviewer #1: No: I did not get access to the data, but the authors stated in the manuscript that data can be provided upon request.

Reviewer #2: Yes

PLOS authors have the option to publish the peer review history of their article (what does this mean?). If published, this will include your full peer review and any attached files.

Reviewer #1: Yes: Laura Fontanesi

Reviewer #2: No

---

## [Decision Letter · Decision Letter 1]

11 Nov 2019

Dear Dr Peters,

Thank you very much for submitting your manuscript, 'The drift diffusion model as the choice rule in inter-temporal and risky choice: a case study in medial orbitofrontal cortex lesion patients and controls.', to PLOS Computational Biology. As with all papers submitted to the journal, yours was fully evaluated by the PLOS Computational Biology editorial team, and in this case, by independent peer reviewers. The reviewers appreciated the attention to an important topic but identified some aspects of the manuscript that should be improved.

As you can see below, one of the reviewers had a small, but reasonable, request regarding the prior of one of the parameters, which should be easily addressed.

While the data underlying the study can not be made publically available, we would encourage you to also find a secondary place for storing data, e.g. an institutional data access portal, should your institution have this. 

We would  like to ask you to modify the manuscript according to the review recommendations before we can consider your manuscript for acceptance. Your revisions should address the specific points made by each reviewer and we encourage you to respond to particular issues Please note while forming your response, if your article is accepted, you may have the opportunity to make the peer review history publicly available. The record will include editor decision letters (with reviews) and your responses to reviewer comments. If eligible, we will contact you to opt in or out.raised.

- Supporting Information uploaded as separate files, titled 'Dataset', 'Figure', 'Table', 'Text', 'Protocol', 'Audio', or 'Video'.

We hope to receive your revised manuscript within the next 30 days. If you anticipate any delay in its return, we ask that you let us know the expected resubmission date by email at ploscompbiol@plos.org.

Sincerely,

Ulrik R. Beierholm

Associate Editor

PLOS Computational Biology

Samuel Gershman

Deputy Editor

PLOS Computational Biology

[LINK]

Reviewer's Responses to Questions

**Comments to the Authors:**

Reviewer #1: Thank you for the additional mixed-models analyses. I only have one question about that, and it’s why was the lambda parameter constrained to be [-3, 3] in the standard normal space. Judging by the posteriors, it looks like the distributions are all pushing towards the bound at 3. If possible, I would consider relaxing the prior distribution to accommodate higher values of lambda.

I understand why it’s not feasible to fit separate DDM parameter by condition.

I also appreciate the change in wording regarding the prior distributions.

Thanks for the parameter recovery. It looks like the Bayesian estimation procedure is indeed able to recover parameters well.

Posterior predictive are also much improved. I really like how the also provide a better explanation of why the non-linear mapping fits the data better.

Regarding the Cohen’s d calculation, I simply suggested to perform a calculation based on all posterior samples and not just on their summaries (e.g., the mean). So that the result would then be a Cohen’s d distribution, instead of just a point summary. But I understand that this is not a crucial part of the results and this should be enough for future reference purposes.

I appreciate that the correlation analyses between raw data and DDM parameters are now out of the main text.

Reviewer #2: I would like to thank the authors for revising the manuscript and providing a detailed response. I believe this manuscript, and mainly the demonstration that a nonlin RLDDM can provide a better fit to choice&RT value-based data, serves as an important contribution to the field.

All the best,

Nitzan Shahar.

**Have all data underlying the figures and results presented in the manuscript been provided?**

Reviewer #1: Yes

Reviewer #2: No: Authors mention data will be provided on-demand.

PLOS authors have the option to publish the peer review history of their article (what does this mean?). If published, this will include your full peer review and any attached files.

Reviewer #1: No

Reviewer #2: No

---

## [Editor Report · Decision Letter 2]

19 Dec 2019

Dear Dr Peters,

We are pleased to inform you that your manuscript 'The drift diffusion model as the choice rule in inter-temporal and risky choice: a case study in medial orbitofrontal cortex lesion patients and controls.' has been provisionally accepted for publication in PLOS Computational Biology.

In the meantime, please log into Editorial Manager at https://www.editorialmanager.com/pcompbiol/, click the "Update My Information" link at the top of the page, and update your user information to ensure an efficient production and billing process.

One of the goals of PLOS is to make science accessible to educators and the public. PLOS staff issue occasional press releases and make early versions of PLOS Computational Biology articles available to science writers and journalists. PLOS staff also collaborate with Communication and Public Information Offices and would be happy to work with the relevant people at your institution or funding agency. If your institution or funding agency is interested in promoting your findings, please ask them to coordinate their releases with PLOS (contact ploscompbiol@plos.org).

Thank you again for supporting Open Access publishing. We look forward to publishing your paper in PLOS Computational Biology.

Sincerely,

Ulrik R. Beierholm

Associate Editor

PLOS Computational Biology

Samuel Gershman

Deputy Editor

PLOS Computational Biology

---

## [Editor Report · Acceptance letter]

15 Apr 2020

PCOMPBIOL-D-19-01092R2 

The drift diffusion model as the choice rule in inter-temporal and risky choice: a case study in medial orbitofrontal cortex lesion patients and controls

Dear Dr Peters,

I am pleased to inform you that your manuscript has been formally accepted for publication in PLOS Computational Biology. Your manuscript is now with our production department and you will be notified of the publication date in due course.

With kind regards,

Matt Lyles
